# Specific Domains of Creativity and Their Relationship with Intelligence: A Study in Primary Education

Olivia López-Martínez [1,*] and Antonio José Lorca Garrido [2]

1   Department of Evolutionary and Educational Psychology, University of Murcia, 30100 Murcia, Spain
2   Faculty of Education, University of Murcia, 30100 Murcia, Spain; antoniojose.lorca1@um.es
*   Correspondence: olivia@um.es

**Abstract:** This study focuses on the relationships between creativity and intelligence. The main objectives of this study are to know the relationship between creativity and intelligence manifested in individuals aged 9 to 12 and to predict the creativity scores from IQ scores. The design of this study is non-experimental with a correlational, cross-sectional, quantitative approach. In order to achieve the stated objectives, several education centres located in Murcia were selected, in which 323 students took part in a creativity test (PIC-N) and a test about intelligence, depending on the educational level (BADYG/E2r or BADYG/E3r). The results obtained were that intelligence (IQ) was related to general creativity and narrative creativity, but not to the specific domain of graphic creativity. In addition, the analyses indicate that scores on general creativity ($F_{(1,321)} = 14.302$, $p < 0.01$) and narrative creativity ($F_{(1,321)} = 14.114$, $p < 0.01$) can be predicted from the IQ. At the educational level, language is a determining factor in narrative creativity and, in turn, language is consolidated as children's cognitive development proceeds.

**Keywords:** general creativity; narrative creativity; graphic creativity; intelligence; primary education





## 1. Introduction

Creativity is one of the social demands of the 21st century, and the improvement of creative thinking is a competence that is at the level of literacy or digitalization [1]. Creativity and intelligence have several factors in common, such as knowledge, memory, divergent production, convergent production and evaluation [2]. European education systems focus on the development of intelligence as the dominant ability, although the need to work on creativity is increasing [3].

Intelligence is a context-dependent element requiring goal-oriented effectiveness, while the dynamic definition of creativity is assumed as a context-related element (cultural and social environment) requiring potential originality and effectiveness [4]. Similarly, we understand creativity as the capacity to generate many ideas on a particular topic, and is therefore intrinsically related to divergent thinking [5].

The relationships between creativity and intelligence have been studied for years although there is no consensus on them. This controversy is due to the great amount of empirical studies on this subject that have been contradictory, because there are different strategies to evaluate creativity based on different definitions of it [6]. Moreover, the performance of individuals in creativity tests is as important as their ability to interpret them [7].

One of the main areas of research into creativity is the relationship of intelligence to divergent thinking [8]. Therefore, intelligence is discussed in a study on creativity because in order to achieve creative achievement, intelligence needs to be put into action to convert competencies into remarkable creative achievements [9].

The subjects have to make decisions to achieve the best result so both variables come into play [10]. Thanks to creativity, they can adapt to changes and generate solutions to problems [11]. However, in order to do this, it is essential to build and use spaces of

freedom and proposals in the classroom that allow creativity to appear when interacting with the context [12], especially in the first years of schooling [13].

Certain minimum levels of intelligence are necessary for creative capacity to appear at different levels [14]. This would mean that it would be impossible for creativity to exist in any individual in the absence of intelligence. Much of the research is along these lines as it supports the threshold theory [15], which implies that, in order to be intelligent, it is necessary to have a medium-high level of intelligence [16]. However, there is no convincing evidence to support the threshold theory and it is very complicated to set concrete limits on it [5]. These arguments that support or oppose the threshold theory show that there is no universal consensus.

The current trend in creativity is towards domain-specific creativity due to the low correlations found in the creative product in the different domains of creativity [17]. Therefore, a subject who is creative in a certain field, such as drawing, may not be creative in writing a text [18,19]. General creativity is the sum of creativity in its narrative and graphic domain [20]. Narrative creativity is the value of divergent thinking in the resolution of problems with verbal content, while graphic creativity is applied to non-verbal problems [21].

Intelligence as well as creativity are capacities that individuals have that are often brought to light by interaction in different contexts that are mostly due to external conditions [22]. An external situation that requires the resolution of a problem could lead to the appearance of intelligence or creativity to deal with it, even the appearance of both since they are not incompatible. Intelligence is a capacity that is often valued by the society in which we live, and which is always attributed a positive connotation but is not always used in the right way [23]. As with creativity, intelligence can also be used to do evil, so in both lies a high potential for carrying out differentiating actions, which can cause both good and evil.

O'Hara and Sternberg [24] establish that there are five perspectives to deal with the relationship between creativity and intelligence. The first maintains that creativity is an element of intelligence, as established in Guilford's Theory [25]. The second defines intelligence as an element of creativity, as specified in the studies of Sternberg and Lubart [18]. The third approach establishes that, in some cases, intelligence and creativity are similar while in other cases they are different. The fourth position identifies creativity and intelligence as overlapping elements. The last position argues that intelligence and creativity are different constructs and are therefore independent of each other.

In Guilford's Theory [2], both divergent and convergent thinking are included, meaning intelligence is essential in the emergence of creative achievement. Creativity means to propitiate ideas suitable to the task that are original and elaborated, considering the use of convergent thinking to be able to select the idea or ideas that we are going to use in a certain situation in which we are immersed [19]. Thus, providing an unexpected response to a problem that allows its resolution in its right measure [14].

The relationship between creativity and human intelligence is clearer than it might seem, since the latter is characteristic of the human condition, creative and efficient, and can be developed through education [26]. This definition of intelligence could be included within the first perspective [24], since it shows that creativity is a part of intelligence.

Many of the investigations into the possible relations between creativity and intelligence [22,27–30] point to the existence of low positive correlations between creativity and intelligence, taking the IQ as an indicator.

As detailed above, the relationship between creativity and intelligence can be interpreted in different ways. These constructs show several similarities, especially at the school stage, as well as in their behaviour according to the gender variable. Regarding differences between male and female patterns of creativity, a number of studies show that there is no clear relationship between creativity and gender [31–33]. The functional magnetic resonance techniques showed that there was no difference between the creative performance of men and women [34]. Studies in school children also showed that boys and girls showed

similar levels of creativity [35], establishing that there is a null relationship between the creativity and gender of the participants.

On the other hand, there are numerous studies that have analysed the differences based on the gender of the participants in terms of intelligence. A review of the existing literature on this subject states that the differences in early ages are small or inexistent and that it is from adolescence and especially in adulthood when these differences are more noticeable [36]. In the school period, there are no significant differences by gender in the stage from 6 to 14 years old, with these differences only starting to appear in subjects who are older than 15 years old [37].

Taking as a reference the previously exposed theoretical approaches and studies, the indications of the Ortiz-Ocaña publication were taken into account for the writing of the objectives of the present study [38]. The general objectives of the research were to know the relationship between creativity and intelligence from individuals between 9 and 12 years old and predict the creativity scores from IQ scores, with the purpose of adopting as valid the third approach proposed by O'Hara and Sternberg [24]. Likewise, the hypothesis was raised that creativity is related with intelligence and another hypothesis was that IQ scores predict creativity scores. To address this approach, the following specific objectives were previously addressed:

1.  Investigate the performance of the verbal factor, the numerical factor and the spatial factor of the participants.
2.  Know the levels of creativity and intelligence in subjects from 9 to 12 years old.
3.  Contrast whether there are differences in the levels of creativity and intelligence according to the gender of the participants.
4.  Compare the levels of creativity and intelligence according to the level of education to check whether there are significant differences.

## 2. Materials and Methods

The design of the present study is of a non-experimental, quantitative nature, since it aims to show the relationship between certain variables that are not intentionally manipulated, only the phenomena are observed in their natural environment to analyse them [39]. Depending on its sequence or temporal dimension, this study is transversal or transsectional, since the data collection is done in a single moment in time [40]. The scope of this research is correlational in that it aims to find out the relationship or degree of association that exists between two or more variables in a simple [41].

In total, 323 subjects in primary education, registered in primary schools in the Region of Murcia in Spain, participated. The families of the students in this study are of low socio-economic status. The individuals belong to the lowest educational levels, being 103 students in the fourth grade of primary school (31.93%), 111 in the fifth grade of primary school (34.45%) and 109 in the sixth grade of primary school (33.75%). Therefore, they were between the ages of 9 and 12. In terms of gender, 168 boys (52.01%) and 155 girls (47.99%) took part. The participants were selected from a consecutive non-probability sampling, which was deemed comprehensive, since all individuals from the educational levels, previously selected according to the criteria of the researcher, that belonged to the last stretch of primary education were taken into account [42].

The data were obtained by applying three standardized, validated and reliable tests to obtain information about the variables in this study.

"Prueba de Imaginacion Creativa para niños (PIC-N)" [20]: this test allows to obtain a general creativity score from the students' performance in two specific domains, such as narrative and graphic creativity [20]. It is designed for individuals between the ages of 8 and 12. The PIC-N is made up of four games. The first three games assess narrative creativity but the first one does not consider the originality variable [20]. In game one, individuals have to write down everything that is happening in an image of a child opening a chest [20]. In game two, subjects have to identify all possible uses of a rubber tube, of which they have seven, and in game three, individuals have to describe what would happen

if all squirrels turned into dinosaurs [20]. In the last game, subjects have to draw four pictures from given strokes and assign a title to each of the drawings in order to evaluate graphic creativity [20]. With regard to the reliability of the test, the Cronbach's alpha value was very satisfactory ($\alpha = 0.84$).

"Bateria de Aptitudes Diferenciales y Generales E2 renovado (BADYG/E2r)" [43]: this test allows the measurement of the intelligence construct from the sum of three factors, namely, the verbal factor, numerical factor and spatial factor, in third- and fourth-year primary school students [43]. The test consists of six tests (two of each factor): analogical relations, numerical problems, logical matrices, sentence completion, numerical calculation and rotated figures. Each of these tests is made up of 24 elements; therefore, the maximum score in each of them is 24. By adding up the results of all the tests, a general intelligence score is obtained, which together with the date of birth allows an approximation of the IQ. The Cronbach's alpha for this instrument was 0.79.

"Bateria de Aptitudes Diferenciales y Generales E3 renovado (BADYG/E3r)" [44]: this test allows the measurement of the intelligence construct from the sum of three factors, namely, the verbal factor, numerical factor and spatial factor in students in the fifth and sixth year of primary education, as well as in the first year of secondary school [44]. The test consists of six tests (two of each factor): verbal analogies, numerical series, logical matrices, sentence completion, numerical problems and figure fitting. Each of these tests is made up of 32 elements, so the maximum score in each of them is 32. Adding up the results of all the tests gives a general intelligence score, which, together with the date of birth, allows an approximation of the IQ. With regard to the reliability of the test, a very satisfactory Cronbach's alpha of 0.88 was obtained.

First of all, school approval was sought, and parental consent was provided for their children to participate in the study. Parental consent obtained, the instruments relating to intelligence were administered according to educational level, with the BADYG/E2r level for the fourth-year students and the BADYG/E3r level for the fifth- and sixth-year students. These were carried out in the classroom collectively and took approximately one hour and 15 min.

In the following days, the information collection instrument on creativity (PIC-N) was applied to all individuals for approximately 40 min. Similarly, this test was administered in the corresponding classrooms of the subjects collectively. The correction and interpretation of the information collected in the instruments was carried out according to the guidelines established in the corresponding manuals.

A database was created in IBM SPSS Statistics 24.0 statistical analysis software [45]. The information, collected with the instruments, was emptied into this database to develop the statistical analyses that would allow the study's objectives to be met:

1. First, descriptive statistics were used to show information about the verbal factor, the numerical factor, the spatial factor, the IQ, narrative creativity, graphic creativity and general creativity.
2. Then, the Kolmogorov–Smirnov normality test was applied to determine the relevance of the use of parametric tests. As the sample met the parametric requirements, repeated measures ANOVA was used to establish significant differences between the intelligence factors. The independent samples *t*-test was used to determine if there were significant gender differences in the variables. Similarly, in order to contrast whether significant differences exist according to the educational level of the individuals, the contrast of means one-way ANOVA was used.
3. Once a descriptive and inferential analysis was carried out, a simple linear regression was used to predict the creativity scores and to find out the relationship between creativity and intelligence.

## 3. Results

This section analyses the data obtained from the different instruments for collecting information. With regard to the specific objective, number one was investigating the

performance of the verbal factor, numerical factor and spatial factor of the participants. In the fourth class, where the information comes from the BADYG/E2r, the score obtained by the students in the numerical factor is the highest of the three factors, while the score of the spatial factor is the lowest. The verbal factor score lies between the two. As can be seen in Table 1, the differences are significant between factors. However, it is necessary to apply the Bonferroni pairwise comparison to find out which factors are significant. This statistic shows that there are significant differences between the numerical and spatial factors ($p < 0.01$), and between the numerical and verbal factors ($p < 0.01$).

**Table 1.** Factors of intelligence.

| Instruments | Verbal Factor | | Numerical Factor | | Spatial Factor | | F | *p* |
|---|---|---|---|---|---|---|---|---|
| | M (SD) | K-S | M (SD) | K-S | M (SD) | K-S | | |
| BADYG/E2r | 25.74 (7.74) | 0.2 | 31.5 (9.16) | 0.2 | 25.26 (6.34) | 0.2 | 12.118 | 0.000 ** |
| BADYG/E3r | 27.9 (11.39) | 0.2 | 26.02 (13.21) | 0.2 | 26.09 (11.39) | 0.2 | 1.855 | 0.163 |

Note: K-S = bilateral sig. of Kolmogorov–Smirnov test. ** $p < 0.01$.

On the other hand, the information for the fifth and sixth grades comes from the BADYG/E3r. The score obtained by students in the verbal factor is the highest of the three factors, while the score of the numerical factor is the lowest. However, the numerical factor score is very close to the spatial factor. Nevertheless, these small differences in the levels of these factors are not significant, as indicated by the ANOVA.

Regarding the second specific objective, we wanted to know the levels of creativity and intelligence in subjects from 9 to 12 years old. Three types of creativity are considered: narrative creativity ($M = 56.29$; $SD = 24.76$) presents high levels in comparison with graphic creativity ($M = 11.67$; $SD = 4.17$), being in both cases heterogeneous groups. General creativity ($M = 67.97$; $SD = 26.09$) is the sum of the two previous ones; therefore, higher levels are obtained in a group that is also heterogeneous. Regarding intelligence, the score is given by the IQ ($M = 90.28$; $SD = 16.71$).

Before tackling the inferential analysis, it is necessary to ensure that the variables are distributed according to the normal distribution. For this purpose, Kolmogorov–Smirnov's normality test was used, which gave the following results: narrative creativity ($p > 0.05$), graphic creativity ($p > 0.05$), general creativity ($p > 0.05$) and intelligence ($p > 0.05$). Therefore, all the variables are distributed following the criterion of normality.

To address the third specific objective, an independent samples *t*-test was carried out in order to contrast whether there are differences in the levels of creativity and intelligence according to the gender of the participants (Table 2).

**Table 2.** Creativity and intelligence according to the gender of the participants.

| Variables | Male M (SD) | Female M (SD) | t | *p* |
|---|---|---|---|---|
| General creativity | 54.90 (25.32) | 57.81 (24.27) | −0.637 | 0.525 |
| Narrative creativity | 11.69 (4.01) | 11.65 (4.37) | 0.058 | 0.954 |
| Graphic creativity | 66.60 (26.51) | 69.46 (25.78) | −0.596 | 0.553 |
| IQ | 91.31 (17.91) | 89.16 (15.37) | 0.700 | 0.485 |

As shown in the table above, the *t*-test of narrative creativity (t = −0.637, df = 321, $p > 0.05$) shows that there are no significant differences between the levels of narrative creativity of the male and female gender. Similarly, in graphic creativity (t = 0.058, df = 321, $p > 0.05$) and in general creativity (t = −0.596, df = 321, $p > 0.05$) there are also no significant differences according to the gender of the participants. The IQ test T scores (t = 0.70, df = 321, $p > 0.05$) are on the same line. In short, there are no significant differences according to gender in any of the variables of this study.

With regard to specific objective number four—comparing the levels of creativity and intelligence according to the educational level, to check whether significant differences exist—it was necessary to use a one-way ANOVA (Table 3) to reach this objective.

**Table 3.** Creativity and intelligence according to the educational level of the participants.

| Variables | Fourth M (SD) | Fifth M (SD) | Sixth M (SD) | F | p |
|---|---|---|---|---|---|
| Narrative creativity | 52.53 (24.13) | 59.83 (24.24) | 56.25 (25.94) | 0.856 | 0.428 |
| Graphic creativity | 13.39 (3.85) | 10.98 (4.34) | 10.75 (3.86) | 5.138 | 0.007 ** |
| General creativity | 65.92 (25.82) | 70.80 (25.36) | 67 (27.46) | 0.383 | 0.683 |
| IQ | 98.20 (12.65) | 84.99 (17.11) | 88.16 (17.25) | 7.358 | 0.001 ** |

** $p < 0.01$.

As shown in the table above, ANOVA shows that there are no significant differences according to educational level in narrative creativity (F (2,320) = 0.856, $p > 0.05$) and in general creativity (F (2,320) = 0.383, $p > 0.05$). However, ANOVA results on graphic creativity (F (2,320) = 5.138, $p < 0.01$) indicate that there are significant differences according to educational level in graphic creativity. The Tukey post hoc test shows that significant differences ($p < 0.05$) occurred between the fourth and fifth primary grades, with a mean difference of 2.42. There also are significant differences ($p < 0.05$) between the fourth and sixth primary grades, with a mean difference of 2.65.

On the other hand, the ANOVA results of the IQ (F (2,320) = 7.358, $p < 0.01$) show that there are significant differences in IQ according to the educational level. The Tukey post hoc test shows that significant differences ($p < 0.01$) occurred between the fourth and fifth primary grades, with a difference in means of 13.20. In turn, there are significant differences ($p < 0.05$) between the fourth and sixth primary grades, with a mean difference of 10.04.

Finally, to answer the general objective, which is to know the relationship between creativity and intelligence from individuals between 9 and 12 years old and to predict the creativity scores from IQ scores, simple linear regression was used, which showed the following results (Table 4).

**Table 4.** Simple linear regression model—the influence of IQ on different types of creativity.

| Kind of Creativity | $R^2$ | F (p) | Beta (ET) | B | t (p) |
|---|---|---|---|---|---|
| General creativity | 0.109 | 14.302 (0.00) ** | 0.330 | 0.515 | 3.782 (0.00) ** |
| Narrative creativity | 0.108 | 14.114 (0.00) ** | 0.328 | 0.486 | 3.757 (0.00) ** |
| Graphic creativity | 0.014 | 1.620 (0.206) | 0.117 | 0.029 | 1.273 (0.206) |

** $p < 0.01$.

As the results in the table above indicate, the graphic creativity scores (F (1,321) = 1.620, $p > 0.05$) cannot be predicted from the IQ scores. However, the scores for general creativity (F (1,321) = 14.302, $p < 0.01$) and narrative creativity (F (1,321) = 14.114, $p < 0.01$) showed that both models are statistically significant. Therefore, the first model can be predicted from the IQ scores in 10.9% of the cases, while the second model only can be predicted from the IQ scores in 10.8% of the cases. The participants' predicted general creativity is equal to 21.432 + 0.515 (IQ) and their predicted narrative creativity is equal to 12.391 + 0.486 (IQ). The hypothesis that IQ scores predict creativity scores is fulfilled but only in a general and narrative way.

## 4. Discussion

The first specific objective was to investigate the performance of the verbal factor, numerical factor and spatial factor of the participants. Taking into account the results of the research, it was found that in the fourth year of primary education the highest scores were

obtained in the numerical factor, followed by the scores of the verbal factor and finally the spatial factor. The description of the fourth primary school year curriculum establishes mathematics and deepening mathematics as subjects to work on the numerical factor, so it is the highest, with significant differences with respect to the verbal factor and the spatial factor. However, the highest scores in the primary fifth and sixth grades were for the verbal factor, followed by the spatial factor and the numerical factor scores, which differed only a few hundredths of a point from each other but there were no significant differences. The very low results in the spatial factor, although not significant, may be due to it not usually being worked on at school compared to the other factors [43,44]. The spatial factor is not governed by cultural aspects compared to the numerical and verbal factors, so it is more free [43,44].

With regard to the second specific objective of knowing the levels of creativity and intelligence in subjects from 9 to 12 years old, in view of the results, it can be seen that individuals obtain scores in the three types of creativity that fit the composition of the PIC-N [20]; therefore, the scores ordered from highest to lowest were general creativity, narrative creativity and graphic creativity. In terms of IQ, the average levels are almost 10 points below 100, which is the average of the normal distribution of the IQ, meaning the sample mean is below the mean of the standardised sample [43,44]. These low levels of intelligence may be due to the social and economic context of the schools.

With reference to the third specific objective of contrasting whether there are differences in the levels of creativity and intelligence according to the gender of the participants, given the results of this study, no significant differences were found between the different types of creativity and the gender of the participants; therefore, gender is not a determining variable in the levels of creativity presented by the participants [31–35]. Likewise, intelligence does not present significant differences according to the gender of the participants, which is due to the fact that individuals from 6 to 14 years old do not present differences according to gender [36,37].

The fourth specific objective was to compare the levels of creativity and intelligence according to educational level, to check whether there are significant differences. In graphic creativity, it is clear that individuals are more creative at a younger age, implying that as they grow older their graphic creativity decreases. However, this is not the case for general creativity and narrative creativity since the levels they present are similar in the three educational levels. In intelligence (IQ), significant differences were found depending on the educational level between the fourth and fifth primary grades, and between the fourth and sixth primary grades. In view of the results, it can be deduced that individuals in the fourth year of primary school have comparatively better performance than the rest of the students in later years. Similarly, it can be observed that the fourth year of primary education is a more homogeneous group compared to the fifth- and sixth-year groups, so that the scores of the latter are more dispersed due to the fact that they are more heterogeneous groups.

In terms of the general objective of relating creativity with intelligence and predicting creativity scores from IQ scores, it can be stated, in view of the results obtained in the simple linear regression model, that there is a relationship between the general creativity and intelligence variables. This relationship between creativity and intelligence occurs in general and in the specific narrative domain. However, this relationship does not occur within the graphic domain of creativity. The results described about the relationship between creativity and intelligence are similar to the results of many studies [22,27–30].

Similarly, the results of the statistical analysis show that intelligence acts as a predictor of narrative creativity and general creativity [22,27–30]. However, IQ does not act as a predictor of graphic creativity; therefore, intelligence is not as relevant in this type of creativity as it is in narrative creativity. In view of the results of the regression analysis, assume that the relationship is linear and that there is in fact no threshold [45], although it is worth remembering that this approach is not universally accepted.

In educational practice, we can see a greater relevance of intelligence in narrative creativity, as language is a determining factor in this type of creativity and, in turn, language

is consolidated as children's cognitive development takes place. However, schoolchildren's drawing, in which graphic creativity is a determining factor, depends to a certain extent on fine psychomotor skills and less on intelligence.

The main limitation of this study was the impact of COVID-19, as it prevented the study sample from being larger due to the impossibility of accessing more educational centres. COVID-19 increases the difficulties encountered in carrying out research, especially in those cases where access is needed to a child population in which the instruments cannot be supplied by telematic means. Another one of the limitations of the present study was that there are variables that cannot be controlled, such as the fatigue caused by individuals when carrying out these tests. In this study, the mean IQ is almost 10 points below the mean in the participants, so one of the limitations is to extend these results to a normal population. However, they could be extended to contexts that reproduce the same conditions. Finally, as this was a study with schoolchildren, the sample was not chosen randomly.

## 5. Conclusions

The main conclusion derived from this study is that, after taking into account the results, we opt for the third approach of O'Hara and Sternberg [24], which establishes that, in some cases, intelligence and creativity are similar while in other cases they are different. As we can see, general creativity and narrative creativity are similar in relation to intelligence. On the other hand, in the case of graphic creativity, it is different in relation to intelligence. However, the debate on the relationship between creativity and intelligence is still open, although we have shown an approach from the specific domains of creativity [22].

The future objectives derived from this research are the carrying out of more studies within the specific domains of creativity, studies which would be able to relate creativity to intelligence, taking other measures of intelligence, such as those based on the Catell–Horn–Carroll model [46], and carrying out the same study with specific populations, such as individuals with high abilities or individuals with disabilities.

**Author Contributions:** Conceptualization, O.L.-M. and A.J.L.G.; methodology, O.L.-M. and A.J.L.G.; software, O.L.-M. and A.J.L.G.; validation, O.L.-M. and A.J.L.G.; formal analysis, O.L.-M. and A.J.L.G.; investigation, O.L.-M. and A.J.L.G.; resources, O.L.-M. and A.J.L.G.; data curation, O.L.-M. and A.J.L.G.; writing—original draft preparation, O.L.-M. and A.J.L.G.; writing—review and editing, O.L.-M. and A.J.L.G.; visualization, O.L.-M. and A.J.L.G.; supervision, O.L.-M. and A.J.L.G. All authors have read and agreed to the published version of the manuscript.

**Funding:** This research received no external funding.

**Institutional Review Board Statement:** Not applicable.

**Informed Consent Statement:** Not applicable.

**Data Availability Statement:** The data presented in this study are available on request from the corresponding author.

**Acknowledgments:** The authors thank both students and schools for their participation and collaboration in this study.

**Conflicts of Interest:** The authors declare no conflict of interest.

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
