# Peer review of "Specific Domains of Creativity and Their Relationship with Intelligence: A Study in Primary Education"

_sustainability, doi:10.3390/su13084228_

Round 1

Reviewer 1 Report

This research work focuses on the relationships between creativity and intelligence. The main objectives of this study are to know the relationship between creativity and intelligence in 12-year-old students and to predict creativity scores from IQ scores.

Overall, the document is well written and well founded. The research design is correct and the data that have been used seem useful for the objectives that the author of the text proposes.

Like Picasso maintained that inspiration exists, but it has to find you working, this article starts from the premise that creativity exists, but a certain degree of intelligence is necessary.

In this research work, it is established as the first general conclusion that, in view of the results, the third approach of O'Hara and Sternberg is chosen - which establishes that, in some cases, intelligence and creativity are similar while in others cases are different. I believe that the objectives should have specified this "commitment" to the O'Hara and Sternberg approach as one of the main objectives.

On the other hand, although I find the result interesting in that IQ does not act as a predictor of graphic creativity, so intelligence is not as relevant in this type of creativity as it is in narrative creativity; further reflection on the implications that this entails in educational practice would be lacking.

Finally, in the section corresponding to future lines of research, we are invited to carry out more studies with the specific domains of creativity, but taking "other measures of intelligence" as a reference. It would be good for the author to propose to us what other intelligence measures he considers necessary or pertinent.

Author Response

Dear Reviewer

Thank you for your suggestions and improvements to the work. We are now proceeding to a major revision with the reviewers' comments. This additional work will hopefully be considered for the manuscript to be accepted. All changes have been incorporated through the change control and in each of the suggestions below you are referred to the lines or paragraphs in the manuscript where these changes are to be found.

Best regards.

Introduction: -There is limited justification as to why the authors chose to investigate the relationships between creativity and intelligence and what the previous literature says about this relationship. -The Introduction provides a brief overview of key literature on the topic of the relationship between creativity and intelligence. However, key constructs such as creativity, intelligence and divergent thinking are not defined clearly. The discussion of the threshold theory is somewhat simplistic and could be more nuanced.

Intelligence is a context-dependent element requiring goal-oriented effectiveness, while the dynamic definition of creativity is assumed as a context-related element (cultural and social environment) requiring potential originality and effectiveness [4]. Similarly, we understand creativity as the ability to produce divergent ideas; therefore, we do not differentiate between creativity and divergent thinking.

View from line 30 to 34 of the manuscript

For example, the theory is presented in a fairly uncritical manner and appears to suggest that threshold theory applies to all types of creativity and intelligence. The discussion could therefore have touched upon the issue of how creativity is assessed/scored and the impact this has on the relationship with intelligence (see e.g. Jauk et al., 2013).

Certain minimum levels of intelligence are necessary for creative capacity to appear at different levels [14]. This would mean that it would be impossible for creativity to exist in any individual in the absence of intelligence. Much of the research is along these lines as it supports the threshold theory [15] which implies that in order to be intelligent it is necessary to have a medium-high level of intelligence [16]. However, no convincing evi-dence for the existence of the threshold has been found; these limits are arbitrary, and which leaves ample room for degrees of freedom in data analysis in creativity and intelli-gence research.

View from line 49 to 56 of the manuscript.

-In this research work, it is established as the first general conclusion that, in view of the results, the third approach of O'Hara and Sternberg is chosen - which establishes that, in some cases, intelligence and creativity are similar while in others cases are different. I believe that the objectives should have specified this "commitment" to the O'Hara and Sternberg approach as one of the main objectives. -The authors should derive the hypotheses from the literature.

The general objectives of the research were to know the relationship between creativity and intelligence from individuals between nine and 12 years old; and predict the creativity scores from IQ scores, with the purpose of adopting as valid the third approach proposed by O'Hara and Sternberg.

View from line 111 to 112 of the manuscript.

Materials and Methods: -The Materials and Methods section gives a summary of the study, but there could have been more detail on the recruitment of participants (e.g., was parental consent obtained?) and ethics in general.

The scope of this research is correlational in that it aims to find out the relationship or degree of association that exists between two or more variables in a simple

View the seventh paragraph of the materials and methods section. Line 129 to 131.

It would also have been good to see more details on the tests administered, particularly the creativity tasks - although citations are provided for the original tests, these are in Spanish and so are not accessible to many readers.

“Prueba de Imaginacion Creativa para niños (PIC-N)” [20]. The test allows to obtain a general creativity score from the students' performance in two specific domains such as narrative and graphic creativity [20]. It is designed for individuals between the ages of eight and 12. The PIC-N is made up of four games. The first three games assess narrative creativity but the first one does not consider the originality variable [20]. In game one, individuals have to write down everything that is happening in an image of a child opening a chest [20]. In game two, subjects have to identify all possible uses of a rubber tube for which they have seven and in game three, individuals have to describe what would happen if all squirrels turned into dinosaurs [20]. In the last game, subjects have to draw four pictures from given strokes and assign a title to each of the drawings in order to evaluate graphic creativity [20]. With regard to the reliability of the test, Cronbach's alpha value was very satisfactory (α = .84).

Look the fourth paragraph of the materials and methods section. In this paragraph, it describes the creativity test (PIC-N). Line 143 to 154.

-Moreover, the authors should provide an introductory discussion of the setting in which the classrooms are currently carried out and the need to undertake the study carried out. This aims at distinguishing the fundamental assertions guiding the study.

European education systems focus on the development of intelligence as the dominant ability, although the need to work on creativity is increasing [3].

View from line 27 to 29 of the manuscript.

Results: -The Results section follows the planned analyses/objectives, but the analysis could have been more insightful. For example, the threshold theory is accepted without question in the Introduction, yet the correlational and regression analyses assume that the relationship is linear and that there is in fact no "threshold".

In educational practice, we can see a greater relevance of intelligence in narrative creativity, as language is a determining factor in this type of creativity and, in turn, language is consolidated as children's cognitive development takes place. However, schoolchildren's drawing, in which graphic creativity is a determining factor, depends to a certain extent on fine psychomotor skills and less on intelligence.

View from line 325 to 329 of the manuscript.

It is unclear what is added by running both correlation and regression analyses - knowing that intelligence predicts a measure of creativity means that there is a relationship between them.

I agree with you. For this reason, I have removed the correlation analysis as the regression analysis shows that intelligence predicts a measure of creativity means that there is a relationship between them.

The type of post-hoc tests that were conducted.

The Tukey post hoc test shows that significant differences (p < .05) occurred between the fourth of Primary and fifth of Primary with a mean difference of 2.42.

The Tukey post hoc test shows that significant differences (p < .01) occurred between the fourth of Primary and fifth of Primary with a difference in means of 13.20.

View from lines 265 and 274 of the manuscript.

Discussion and conclusions: -The Discussion summarizes the findings, but could go further in interpreting and explaining them. For example, why does the sample have a lower than expected IQ, and what does this mean for the generalizability of the results?

The COVID-19 increases the difficulties encountered in carrying out research, especially in those cases where access is needed to a child population in which the instruments cannot be supplied by telematic means. Another of the limitations of the present study was that there are variables that cannot be controlled, such as the fatigue caused by individuals when carrying out these tests. In this study the mean IQ is almost 10 points below the mean in the participants so one of the limitations is to extend these results to a normal population. However, they could be extended to contexts that reproduce the same conditions. Finally, as this was a study with schoolchildren, the sample was not chosen randomly.

View from line 333 to 336 of the manuscript.

 -The discussion highlights differences between verbal, numerical and spatial factors, concluding that spatial factors are the lowest, but this is based on "eyeballing" the data, without appropriate statistical analysis. Indeed, the differences between spatial and verbal scores is very small (and likely statistically insignificant), yet the discussion suggests otherwise.

Regarding the first specific objective of investigating the performance of the verbal factor, the numerical factor and the spatial factor of the participants. Taking into account the results of the research, it was found in the fourth year of Primary Education that the highest scores were obtained in the numerical factor, followed by the scores of the verbal factor and finally the spatial factor. The description of the Fourth Primary School curricu-lum establishes Mathematics and Deepening mathematics subjects to work on the nu-merical factor, so it is the highest with significant differences with respect to the verbal factor and the spatial factor. However, the highest scores in Primary fifth and sixth were for the verbal factor, followed by the spatial factor and the numerical factor scores which differed only a few hundredths of a point from each other but there are no significant dif-ferences. The very low results in the spatial factor, although not significant, may be due to it is not usually worked on at school like the other factors [43,44]. The spatial factor is not governed by cultural aspects such as the numerical and verbal factors, but is more free [43,44].

View first paragraph of the discussion. Line 275 to 279

-On the other hand, although I find the result interesting in that IQ does not act as a predictor of graphic creativity, so intelligence is not as relevant in this type of creativity as it is in narrative creativity; further reflection on the implications that this entails in educational practice would be lacking.

In educational practice, we can see a greater relevance of intelligence in narrative creativity, as language is a determining factor in this type of creativity and, in turn, language is consolidated as children's cognitive development takes place. However, schoolchildren's drawing, in which graphic creativity is a determining factor, depends to a certain extent on fine psychomotor skills and less on intelligence.

View paragraph before limitations. Line 325 to 329

-The conclusion that "individuals in the fourth year of Primary have greater cognitive abilities than the rest of the students in later years" is misleading - this result compares two different age-adjusted tests and so the authors should emphasize that the younger group have only comparatively better performance and should reflect on the issue of using different intelligence tests.

In terms of the general objective of correlating creativity with intelligence and predicting creativity scores from IQ scores, it can be stated, in view of the results obtained in the simple linear regression model, that there is a relationship between the general creativity and intelligence variables.

View from line 312 to 315 of the manuscript.

-The discussion of limitations and ideas for future research could be more insightful - the sample size was quite large, so this is not really a valid criticism of this study. There could have been more consideration of the validity of the assessment tools and how these affect the generalizability of the findings.

The data collection instruments have been validated by their authors, so their validity is not a problem for generalising the results. Here are the references of the data collection instruments in case you want to check them yourself:

Artola, T.; Ancillo, I.; Mosteiro, P.; Barraca, J. PIC-N. Prueba de Imaginación Creativa para niños; TEA Ediciones: Madrid, España, 2010.

Yuste, C. Batería de aptitudes diferenciales y generales renovado BADYG E2; CEPE: Madrid, España, 2011.

Yuste, C.; Martínez Arias, R.; Galve, J. L. Batería de aptitudes diferenciales y generales renovado BADYG E3; CEPE: Madrid, España, 2011.

-Please include in the methods and study limitations sections the relevant information, underpinning the chosen variables, and study design.

First paragraph of the materials and methods section. The variables chosen, which are general creativity, graphic creativity, narrative creativity, IQ, gender and educational level, are derived from the theoretical approaches in the introduction section. It should even be mentioned that the information collected by the instruments provides the data for the variables.

View line 125 to 131 of the manuscript

Most importantly, please include a more extended version of the limitations of the study both in terms of technique (e.g. underlying assumption) and variables of interest. This suggestion aims to make the manuscript more readable to the international audience. It was not enough to state ‘the COVID-19’ and ‘fatigue’ as the only limitations affecting the results of this study. Also, the study limitations should appear before the conclusions section.

The main limitation of this study was the impact of the COVID-19, as it prevented the study sample from being larger due to the impossibility of accessing more educational centres. The COVID-19 increases the difficulties encountered in carrying out research, es-pecially in those cases where access is needed to a child population in which the instru-ments cannot be supplied by telematic means. Another of the limitations of the present study was that there are variables that cannot be controlled, such as the fatigue caused by individuals when carrying out these tests. In this study the mean IQ is almost 10 points below the mean in the participants so one of the limitations is to extend these results to a normal population. However, they could be extended to contexts that reproduce the same conditions. Finally, as this was a study with schoolchildren, the sample was not chosen randomly.

View the change in the last paragraph of the discussion. Line 330 to 340

Finally, in the section corresponding to future lines of research, we are invited to carry out more studies with the specific domains of creativity but taking "other measures of intelligence" as a reference. It would be good for the author to propose to us what other intelligence measures he considers necessary or pertinent.

The future lines derived from this research are the carrying out of more studies with the specific domains of creativity, studies which relate creativity to intelligence, taking other measures of intelligence based on the Catell-Horn-Carroll model [46] and carrying out the same study with specific populations such as individuals with high abilities or individuals with disabilities.

View line 389 to 394 of the manuscript.

Edición del idioma: en general, a lo largo del artículo, la escritura debe revisarse; hay ejemplos de secciones que no son oraciones y que no tienen mucho sentido. (Por ejemplo, no estaba claro qué querían decir los autores con esta frase. Página 3, línea 122 a 123. 'El alcance de esta investigación es correlacional, ya que la investigación correlacional transversal tiene como objetivo…'.

El alcance de esta investigación es correlacional en el sentido de que tiene como objetivo conocer la relación o grado de asociación que existe entre dos o más variables en un simple [38].

Vea la página 3, línea 138 a 140 del manuscrito.

Reviewer 2 Report

The study dealt with an area of research with little or no attention paid to the literature, methodologies, and the resulted implications. Thus, it was technically unsound, as it lacked methodological rigor in approaching the study. I find the justifications to the problem, methodology, and conclusions sections too problematic to understand. The following points may be helpful for the authors.

  1. There is limited justification as to why the authors chose to investigate the relationships between creativity and intelligence and what the previous literature says about this relationship. It is necessary from the outset to justify how intelligence relates to the components of creativity and the dimensionality of these variables. The authors should derive the hypotheses from the literature. Moreover, the authors should provide an introductory discussion of the setting in which the classrooms are currently carried out and the need to undertake the study carried out. This aims at distinguishing the fundamental assertions guiding the study and avoid possible misinterpretations;
  2. The authors should provide much clearer explanations about the mechanisms and processes applied to customize or adapt the tests to fit the context. How can we trust the items for each category in the absence of, for example, confirmatory factor analysis?
  3. Please include in the methods and study limitations sections the relevant information, underpinning the chosen variables, and study design. Most importantly, please include a more extended version of the limitations of the study both in terms of technique (e.g. underlying assumption) and variables of interest. This suggestion aims to make the manuscript more readable to the international audience. It was not enough to state ‘the COVID-19’ and ‘fatigue’ as the only limitations affecting the results of this study. Also, the study limitations should appear before the conclusions section.
  4. This study has a significant problem in methodology since the authors did not give enough evidence about the relationship patterns, and yet they declared a very tough claim for similarity and differences. As the authors stated on page 7 of the last paragraph, lines 323 to 325,

(The main conclusions derived from this study are, taking into account the results, that we opt for the third approach of O'Hara and Sternberg [28], which establishes that, in some cases, intelligence and creativity are similar while in other cases they are different.)

The fact that IQ predicts creativity does not lead to the conclusion that they are similar or different, as the authors, boldly declared. Instead, it gives evidence that the two variables of interest have some patterns of relationships or not. If the authors wanted to examine the existence of significant differences between the variables, as they boldly claimed in the conclusion, probably they should have first strong literature to their hypothesis that dealt with similarities and differences. Another alternative might be the authors’ use of factor analysis may be to ensure the convergent and divergent nature of the variables of interest. In the case of the authors’ results, they may find evidence of relationship patterns but not similarity and differences. Thus, the final conclusion was not acceptable.

5. Another limitation was the aggregated nature of the IQ tests as these attempt to represent a person's overall intellectual abilities with a single score. As psychologists like Howard Gardner remarked, there are seven distinct forms of intelligence, including musical, kinesthetic, and interpersonal intelligence and others. Given that IQ was measured using aggregate data pauses a serious methodological issue to explain the measures of creativity as data aggregation can result in misleading conclusions. Regression coefficients and their statistical significance differ across degrees of data aggregation. Therefore, the final claim was too problematic. Please read Garrett (2003) to understand why the sign and significance of coefficient estimates from regressions using aggregated data possibly differ from those of regressions that use less aggregated data.

6. The paper benefits from serious language editing. For example, it was not clear what the authors wanted to say with this phrase. Page 3, line 122 to 123. ‘The scope of this research is correlational, since correlational cross-sectional research aims…’

I hope these suggestions are helpful to the author(s).

Author Response

Dear reviewer
Thank you for your comments.
Best regards

Reviewer 3 Report

It is refreshing to see a study on creativity and intelligence that uses a non-English-speaking sample and does not use the conventional creativity and intelligence assessments. As a result, this study has the potential to add to the literature. However, a stronger rationale is needed to justify why it should be published in a journal on sustainability, as opposed to a developmental psychology or creativity journal, for instance. 

The Introduction provides a brief overview of key literature on the topic of the relationship between creativity and intelligence. However, key constructs such as creativity, intelligence and divergent thinking are not defined clearly. The discussion of the threshold theory is somewhat simplistic and could be more nuanced. For example, the theory is presented in a fairly uncritical manner and appears to suggest that threshold theory applies to all types of creativity and intelligence. The discussion could therefore have touched upon the issue of how creativity is assessed/scored and the impact this has on the relationship with intelligence (see e.g. Jauk et al., 2013). The section on gender differences in creativity (or lack thereof) seems out of place and interrupts the flow of the discussion of creativity and intelligence. 

The Materials and Methods section gives a summary of the study, but there could have been more detail on the recruitment of participants (e.g., was parental consent obtained?) and ethics in general. It would also have been good to see more details on the tests administered, particularly the creativity tasks - although citations are provided for the original tests, these are in Spanish and so are not accessible to many readers. 

The Results section follows the planned analyses/objectives, but the analysis could have been more insightful. For example, the threshold theory is accepted without question in the Introduction, yet the correlational and regression analyses assume that the relationship is linear and that there is in fact no "threshold". It is unclear what is added by running both correlation and regression analyses - knowing that intelligence predicts a measure of creativity means that there is a relationship between them. Information from tables is repeated in text unnecessarily, it would be helpful to see df values when reporting t-test and ANOVA results, and the type of post-hoc tests that were conducted. 

The Discussion summarizes the findings, but could go further in interpreting and explaining them. For example, the fact that the average sample IQ was 10 points below the population IQ is stated without any further discussion - why does the sample have a lower than expected IQ, and what does this mean for the generalizability of the results? Elsewhere, differences in the intelligence-creativity relationship are discussed, but not explained - why is there no IQ-graphic creativity relationship when both showed age-related differences? 

The discussion highlights differences between verbal, numerical and spatial factors, concluding that spatial factors are the lowest, but this is based on "eyeballing" the data, without appropriate statistical analysis. Indeed, the differences between spatial and verbal scores is very small (and likely statistically insignificant), yet the discussion suggests otherwise.

The conclusion that "individuals in the fourth year of Primary have greater cognitive abilities than the rest of the students in later years" is misleading - this result compares two different age-adjusted tests and so the authors should emphasise that the younger group have only comparatively better performance, and should reflect on the issue of using different intelligence tests. 

The discussion of limitations and ideas for future research could be more insightful - the sample size was quite large, so this is not really a valid criticism of this study. There could have been more consideration of the validity of the assessment tools and how these affect the generalizability of the findings. 

In general, across the article, the writing needs to be revised - there are examples of non-sentences and sections that don't make complete sense.

Overall, this is a study with potential, but more work is needed for it to achieve that potential.

Author Response

Dear reviewer
Thank you for your comments.
Best regars

Reviewer 4 Report

Dear authors, 

I really enjoyed reading your article and I find it well written, with sound methods, statistical analysis and well-thought discussion. My only suggestions are regarding the formal aspects of the paper, i.e. there are few sentences that (probably by mistake) don´t make that much sense. For exmaple line 67 - there should probably be a comma between those two sentences. Line 317 - it seems there´s something missing in the sentence. SO I would suggest reading the paper once more and correcting the language problems. Other than that, good luck with the publishing process and I hope to see your paper in the journal.

Best regards.

Author Response

Dear reviewer
Thank you for your suggestions. We have already proceeded to make the changes to improve the manuscript.
Best regards

Line 67 - there should probably be a comma between those two sentences.

An external situation that requires the resolution of a problem could lead to the appearance of intelligence or creativity to deal with it, even the appearance of both since they are not incompatible.

View page 2, line 67 of the manuscript.

Line 317 - it seems there´s something missing in the sentence, etc.)

Also, the specific domain of narrative creativity and intelligence present a positive correlation as it did with general creativity.

View line 317.

Round 2

Reviewer 3 Report

Thank you for the updates to your manuscript based on our feedback.

In the previous version, I commented that you appeared to accept the threshold hypothesis unquestioningly, but it seems as if you have gone to the other extreme this time round by saying definitively that there is no evidence for the threshold hypothesis! Although the study you cite is quite convincing, I am not sure that this view is accepted universally, and so I would recommend that this is reflected in your discussion. 

Thank you for clarifying your definitions of key terms. However, your definition for creativity in lines 33-35 appears to contradict what you have written in lines 96-100 - is convergent thinking part of creativity or is it only divergent thinking? 

The Introduction section on gender still feels as though it is out of place, and work is needed to integrate it into the rest of the Introduction. Indeed, the structure of the Introduction could be improved further.

Please note that citations in your added sections are in the wrong format - numerical referencing should be used. 

Thank you for providing further details of the tests in the Material and Methods section - this is very helpful.

The Results section is improved, but there are still some points that need to be addressed from last time (e.g., reporting df values for ANOVAs and t-tests; unnecessary repetition of information from tables in your text). Thank you for clarifying the post-hoc test results; it would also be good to see report the non-significant comparisons for completeness. The reporting of regression results is a little confusing: in your tables, you report B(ET) (beta?) and B, but report the B(ET) value as B in your text. It would be helpful if you could clarify which value is beta (standardized) and B (unstandardized).

Thank you for your changes to the Discussion. I still feel that your conclusion that 4th year children are out-performing children in later years needs more explanation. In the paragraph beginning on line 313, you mention correlation several times, but correlation has been removed from your analysis and so needs to be updated to reflect the focus on regression. In lines 332-333, you state "the mean IQ is almost 10 points below the mean", which is potentially confusing for readers - you will need to clarify that you mean the sample mean is below the standardization sample mean. I still feel you need to consider the validity of the measures before you draw such strong conclusions about "creativity" and "intelligence". 

Writing still needs to revised across the article - there are examples of non-sentences, spelling errors and sections that are unclear.

Author Response

Dear reviewer

Thank you for your suggestions and improvements to the work. We have already proceeded to a major revision with the comments and proposals for improvement suggested by you. All changes have been incorporated through the change control and in each of the suggestions below you are referred to the lines or paragraphs in the manuscript where these changes are found.

Best regards.

In the previous version, I commented that you appeared to accept the threshold hypothesis unquestioningly, but it seems as if you have gone to the other extreme this time round by saying definitively that there is no evidence for the threshold hypothesis! Although the study you cite is quite convincing, I am not sure that this view is accepted universally, and so I would recommend that this is reflected in your discussion.

Similarly, the results of statistical analysis show that intelligence acts as a predictor of narrative creativity and general creativity [22,34-37]. However, IQ does not act as a predictor of graphic creativity, therefore intelligence is not as relevant in this type of creativity as it is in narrative creativity. In view of the results of the regression analysis, assume that the relationship is linear and that there is in fact no threshold [5], although it is worth remembering that this approach is not universally accepted.

View lines 360 to 366

Thank you for clarifying your definitions of key terms. However, your definition for creativity in lines 33-35 appears to contradict what you have written in lines 96-100 - is convergent thinking part of creativity or is it only divergent thinking? 

In this respect we believe that we first give the definition of creativity (lines 33-35) and later explain the five approaches that O'Hara and Sternberg (1999) set out on the different ways of understanding the relationship between creativity and intelligence. What the content in lines 96-100 refers to is one of these approaches.

The Introduction section on gender still feels as though it is out of place, and work is needed to integrate it into the rest of the Introduction. Indeed, the structure of the Introduction could be improved further.

As detailed above, the relationship between creativity and intelligence can be interpreted in different ways. These constructs show several similarities, especially at the school stage, as well as in their behaviour according to the gender variable. Regarding differences between male and female patterns of creativity, a number of studies show that there is no clear relationship between creativity and gender [24-26]. The functional magnetic resonance techniques showed that there was no difference between the creative performance of men and women [27]. Also, studies in school children showed that boys and girls showed similar levels of creativity [28], establishing that there is a null relationship between the creativity and gender of the participants.

On the other hand, there are numerous studies that have analysed the differences based on the gender of the participants in terms of intelligence. The review of the existing literature on this subject states that the differences in early ages are small or inexistent and that it is from adolescence and especially in adulthood when these differences are more noticeable [29]. In the school period, there are no significant differences by gender in the stage from six to 14 years old, starting to appear these differences in subjects who are older than 15 years old [30].

View lines 114 to 129

The Results section is improved, but there are still some points that need to be addressed from last time (e.g., reporting df values for ANOVAs and t-tests; unnecessary repetition of information from tables in your text).

As shown in the table above, the t-test of narrative creativity (t = -.637, df = 321, p > .05) shows that there are no significant differences between the levels of narrative creativity of the male and female gender. Similarly, in graphic creativity (t = .058, df = 321, p > .05) and in general creativity (t = -.596, df = 321, p > .05) there are also no significant differences according to the gender of the participants. The IQ test T scores (t = .70, df = 321, p > .05) are on the same line. In short, there are no significant differences according to gender in any of the variables of this study.

View lines 257 to 263

As shown in the table above, ANOVA shows that there are no significant differences according to educational level in narrative creativity (F (2,320) = .856, p > .05) and in general creativity (F (2,320) = .383, p > .05). However, ANOVA results on graphic creativity (F (2,320) = 5.138, p < .01) indicate that there are significant differences according to educational level in graphic creativity. The Tukey post hoc test shows that significant differences (p < .05) occurred between the fourth of Primary and fifth of Primary with a mean difference of 2.42. Also, there are significant differences (p < .05) between the fourth of Primary and sixth of Primary with a mean difference of 2.65.

On the other hand, the ANOVA results of the IQ (F (2,320) = 7.358, p < .01) show that there are significant differences in the IQ according to the educational level. The Tukey post hoc test shows that significant differences (p < .01) occurred between the fourth of Primary and fifth of Primary with a difference in means of 13.20. In turn, there are significant differences (p < .05) between the fourth of Primary and sixth of Primary with a mean difference of 10.04.

View lines 269 to 282

The reporting of regression results is a little confusing: in your tables, you report B(ET) (beta?) and B, but report the B(ET) value as B in your text. It would be helpful if you could clarify which value is beta (standardized) and B (unstandardized).

As the results in the table above indicate, the graphic creativity scores (F (1,321) = 1.620, p > .05) cannot be predicted from the IQ scores. However, the scores for general creativity (F (1,321) = 14.302, p < .01) and narrative creativity (F (1,321) = 14.114, p < .01) showed that both models are statistically significant. Therefore, the first model can be predicted from the IQ scores in the 10.9 % of the cases, while the second model only can be predicted from the IQ scores in the 10.8 % of the cases. Participants’ predicted general creativity is equal to 21.432 + .515(IQ) and participants’ predicted narrative creativity is equal to 12.391 + .486(IQ). The hypothesis that IQ scores predict creativity scores is fulfilled but only in a general and narrative way.

View lines 289 to 298 and Table 4.

Thank you for your changes to the Discussion. I still feel that your conclusion that 4th year children are out-performing children in later years needs more explanation. 

In intelligence (IQ), significant differences were found depending on the educational level between the fourth of Primary and fifth of Primary, and between the fourth of Primary and sixth of Primary. In view of the results, it can be deduced that individuals in the fourth year of Primary have comparatively better performance than the rest of the students in later years. Similarly, it can be observed that the fourth year of primary education is a more homogeneous group compared to the fifth and sixth year groups, so that the scores of the latter are more dispersed due to the fact that they are more heterogeneous groups.

View lines 342 to 349

In the paragraph beginning on line 313, you mention correlation several times, but correlation has been removed from your analysis and so needs to be updated to reflect the focus on regression.

In terms of the general objective of relating creativity with intelligence and predicting creativity scores from IQ scores, it can be stated, in view of the results obtained in the simple linear regression model, that there is a relationship between the general creativity and intelligence variables. This relationship between creativity and intelligence occurs in general and in the specific narrative domain. However, this relationship does not occur with the graphic domain of creativity. These results described about the relationship between creativity and intelligence are similar to the results of many studies [22,34-37].

View lines 350 to 359

In lines 332-333, you state "the mean IQ is almost 10 points below the mean", which is potentially confusing for readers - you will need to clarify that you mean the sample mean is below the standardization sample mean.

As regards the second specific objective of knowing the levels of creativity and intelligence in subjects from nine to 12 years old. In view of the results, it can be seen that individuals obtain scores in the three types of creativity that fit the composition of the PIC-N [20], therefore, the scores ordered from highest to lowest were: general creativity, narrative creativity and graphic creativity. In terms of IQ, the average levels are almost 10 points below 100 which is the average of the normal distribution of IQ meaning the sample mean is below the mean of the standardised sample [43,44]. These low levels of intelligence may be due to the social and economic context of the schools.

View lines 321 to 328

I still feel you need to consider the validity of the measures before you draw such strong conclusions about "creativity" and "intelligence". 

In this study, standardised tests were administered, the validity of which can be found in the test references:

  1. Artola, T.; Ancillo, I.; Mosteiro, P.; Barraca, J. PIC-N. Prueba de Imaginación Creativa para niños; TEA Ediciones: Madrid, España, 2010.
  2. Yuste, C. Batería de aptitudes diferenciales y generales renovado BADYG E2; CEPE: Madrid, España, 2011.
  3. Yuste, C.; Martínez Arias, R.; Galve, J. L. Batería de aptitudes diferenciales y generales renovado BADYG E3; CEPE: Madrid, España, 2011.

This validity has always been taken into account, together with the reliability of the tests (the latter depending on the sample under study) in order to be able to carry out the research.